# Dead Laying Hens Detection Using TIR-NIR-Depth Images and Deep Learning on a Commercial Farm

**DOI:** 10.3390/ani13111861

**Published:** 2023-06-02

**Authors:** Sheng Luo, Yiming Ma, Feng Jiang, Hongying Wang, Qin Tong, Liangju Wang

**Affiliations:** 1College of Engineering, China Agricultural University, Beijing 100083, China; luosheng@cau.edu.cn (S.L.); myming2020@cau.edu.cn (Y.M.); jiangfeng1@cau.edu.cn (F.J.); hongyingw@cau.edu.cn (H.W.); 2College of Water Resources and Civil Engineering, China Agricultural University, Beijing 100083, China; tongqin@cau.edu.cn

**Keywords:** thermal infrared image, near-infrared image, depth image, image registration, deep learning, dead laying hen detection, large-scale farming

## Abstract

**Simple Summary:**

Timely detection of dead chickens is of great importance on commercial farms. Using multi-source images for dead chicken detection can theoretically achieve higher accuracy and robustness compared with single-source images. In this study, we introduced a pixel-level image registration method to align the near-infrared (NIR), thermal infrared (TIR), and depth images and analyzed the detection performance of models using different source images. The results of the study showed the following: The model with the NIR image performed the best among models with single-source images, and the models with dual-source images performed better than that with single-source images. The model with the TIR-NIR image or the NIR-depth image performed better than the model with the TIR-depth image. The detection performance with the TIR-NIR-Depth image was better than that with single-source images but was not significantly different from that with the TIR-NIR or NIR-depth images. This study provided a reference for selecting and using multi-source images for detecting dead laying hens on commercial farms.

**Abstract:**

In large-scale laying hen farming, timely detection of dead chickens helps prevent cross-infection, disease transmission, and economic loss. Dead chicken detection is still performed manually and is one of the major labor costs on commercial farms. This study proposed a new method for dead chicken detection using multi-source images and deep learning and evaluated the detection performance with different source images. We first introduced a pixel-level image registration method that used depth information to project the near-infrared (NIR) and depth image into the coordinate of the thermal infrared (TIR) image, resulting in registered images. Then, the registered single-source (TIR, NIR, depth), dual-source (TIR-NIR, TIR-depth, NIR-depth), and multi-source (TIR-NIR-depth) images were separately used to train dead chicken detecting models with object detection networks, including YOLOv8n, Deformable DETR, Cascade R-CNN, and TOOD. The results showed that, at an IoU (Intersection over Union) threshold of 0.5, the performance of these models was not entirely the same. Among them, the model using the NIR-depth image and Deformable DETR achieved the best performance, with an average precision (AP) of 99.7% (IoU = 0.5) and a recall of 99.0% (IoU = 0.5). While the IoU threshold increased, we found the following: The model with the NIR image achieved the best performance among models with single-source images, with an AP of 74.4% (IoU = 0.5:0.95) in Deformable DETR. The performance with dual-source images was higher than that with single-source images. The model with the TIR-NIR or NIR-depth image outperformed the model with the TIR-depth image, achieving an AP of 76.3% (IoU = 0.5:0.95) and 75.9% (IoU = 0.5:0.95) in Deformable DETR, respectively. The model with the multi-source image also achieved higher performance than that with single-source images. However, there was no significant improvement compared to the model with the TIR-NIR or NIR-depth image, and the AP of the model with multi-source image was 76.7% (IoU = 0.5:0.95) in Deformable DETR. By analyzing the detection performance with different source images, this study provided a reference for selecting and using multi-source images for detecting dead laying hens on commercial farms.

## 1. Introduction

Eggs are rich in nutrition and indispensable agricultural and livestock products in people’s daily lives [1]. With the application of mechanical and control technology, feed supply, water supply, egg collection, manure removal, and environmental management have been automated in large-scale laying hen farms. However, detecting and removing dead chickens still relies heavily on manual labor [2]. Due to visual fatigue during long-term inspections and difficulty observing chicken cages higher than the worker, some dead chickens may not be found in time, leading to cross-infection, disease transmission, and economic loss. Additionally, prolonged work on a farm with high concentrations of harmful gases and dust harms workers’ health. Therefore, using machines to replace manual inspections and detect dead chickens is increasingly urgent on large-scale commercial farms.

With digital image processing and machine learning development, many scholars have researched dead chicken detection using chicken images. Lu et al. [3] converted chicken RGB images into the L*a*b* color space, extracted a* as a feature, and segmented the red chicken comb. They determined whether a dead chicken was in the cage by judging the presence or absence of a stationary red chicken comb. Zhu et al. [4] and Peng [5] extracted five features from the chicken comb based on the L*a*b* color space and used the variation of these features as input for SVM to determine whether there was a dead chicken in the image. The accuracy of this method in recognizing dead chickens was over 92%. Lu [6] used center radiating vectors to represent chicken contours in images. She used the absolute difference between the corresponding vectors of two contours over a period of time as the input of the support vector machine (SVM) to train the classifier to identify whether an image contains a dead chicken. The classification accuracy of this method could reach 95%. Li [7] collected chicken leg images and counted the shanks to determine the number of chickens in the image and whether there was a dead chicken. The accuracy of this method was greatly affected by chicken density and activity, with an accuracy of 90% in experiments. However, these methods could not detect dead chickens well on commercial farms when the head was obscured or lighting conditions were poor.

Recently, with the development of deep learning, the convolutional neural network (CNN) has been increasingly applied to dead chicken detection, leveraging its powerful feature extraction capabilities. Zhuang et al. [8] proposed a diseased broilers detection network named IFSSD based on the single shot multibox detector (SSD) network, which identified the health status of broilers while detecting them. The mAP (IoU = 0.5) of the network reached 99.7%. Xue [9] established a dead broilers detection model based on the Faster R-CNN algorithm after registering and fusing thermal infrared images with visible light images. This model’s precision and recall were over 98%. Liu et al. [10] designed a small removal system for dead broilers that used the YOLOv4 algorithm to detect dead chickens, with an mAP (IoU = 0.5) of 95.24%. Hao et al. [11] built a dead broiler detection system in a stacked cage environment, including an autonomous detection platform and a dead broiler detection model based on an improved YOLOv3. The experimental results showed that the mAP (IoU = 0.5) of the model was 98.6%.

Despite significant progress made in previous research, several issues still need to be addressed regarding the practical application of dead chicken detection in commercial farming. Firstly, the light intensity is low in commercial farming, with the stacked cage farming system usually below 20 lux. Although visible light cameras contain rich texture information, the imaging efficiency and quality are low under low-light conditions, leading to target loss [12,13]. Using visible light to improve image quality may cause stress on chickens, resulting in reduced production performance or even killing chickens. Secondly, chickens often obstruct each other on commercial farms. The single-source image does not contain enough information about dead chickens due to hardware limitations, leading to problems such as missing, false, repeated, and inaccurate detection [14,15]. Using multi-source images for dead chicken detection is a good solution to solve these issues. Multi-source images generally refer to images from multiple sensors or cameras, including RGB, thermal infrared (TIR), depth, and near-infrared (NIR) images. The NIR image lacks color information compared to the RGB image, but it is not affected by ambient light and still has details and distinct edges [16]. The TIR image contains thermal radiation information emitted by targets. However, it lacks detailed descriptions of the targets and has a low signal-to-noise ratio and contrast [17]. The depth image contains distance information, which can be used for image registration and 3D reconstruction. Nevertheless, the lack of detailed information on object features makes it difficult to identify objects with similar heights and shapes accurately [18].

In the livestock industry, previous research applied multi-source images to different aspects. Liu et al. [19,20,21] developed a registration and fusion algorithm to fuse thermal infrared and visible light images of pigs. They successfully detected the region of the pig ear root using the fused image and an improved active shape model. Zhu et al. [18] proposed an end-to-end pose recognition algorithm for lactating sows using RGB-D images. Their algorithm employed two CNN networks to extract features from the RGB image and depth image separately. Additionally, a region proposal network and a feature fusion layer were utilized to generate regions of interest and merge features of the RGB-D images. The fused features were then input into a Faster R-CNN network to detect the posture of lactating sows. He et al. [22] introduced a Light-weight High-Resolution Network (LiteHRNet) for estimating the weight of sheep using RGB-D images. The LiteHRNet backbone extracted features from the given sheep RGB-D images, and the LEHead module combined these features to estimate the sheep’s weight. Lamping et al. [23] proposed ChickenNet, a network based on the Mask R-CNN architecture. Depth images were added to visible light images in input, and plumage conditions were added to the detection of laying hens in output. Zhang et al. [24] presented a method for monitoring feather damage using RGB, thermal infrared, and depth images. The method utilized the above images as inputs to reconstruct a three-dimensional model of a chicken and established an automated algorithm for assessing the depth of feather damage. With a multi-source image, it is theoretically possible to achieve better detection results than that with a single-source image because it contains different types of information.

This study replaced visible light images with NIR images to avoid ambient light interference and proposed a method of dead laying hen detection using TIR-NIR-Depth images. Firstly, we proposed a pixel-level registration method and projected the NIR image and depth image into the coordinates of the TIR image, resulting in registered images. Then, we used the registered single-source, dual-source, and multi-source images as inputs for object detection networks such as YOLOV8n, Deformable DETR, Cascade R-CNN, and TOOD to analyze the detection performance of each source image. The flow diagram is shown in Figure 1. Overall, this study proposed and evaluated a dead chicken detection method based on multi-source images and deep learning, which helped researchers select and use the multi-source image for hen monitoring on commercial farms.

## 2. Materials and Methods

### 2.1. Image Data Collection

#### 2.1.1. Animals and Farm Environment

All images were collected at a commercial laying hen farm in BeijingHuadu Yukou Poultry Industry Co., Ltd., Beijing, China. The chicken breed was Jingfen 6, about 500 days old. The farm had four rows of chicken cages; each row was divided into three tiers: upper, middle, and lower, and arranged in an A-shaped staircase. Each tier had 300 cages, with four chickens per cage on the upper and middle tiers and three chickens per cage on the lower tier. Each cage was equipped with nipple-type drinkers. The automatic feeding device fed the chickens at 7:00, 11:00, 15:00, and 18:30. The environmental control system regulated the temperature and humidity inside the farm by controlling the fans. Incandescent lamps were used as the light source, with an illumination time of 2:30–18:30 and an illumination intensity range of 5~20 lux.

#### 2.1.2. Image Acquisition Device

Image acquisition device included two cameras, an industrial computer, a self-developed mobile chassis, and some connectors. All images were captured using a depth camera and a TIR camera with fixed relative positions, as shown in Figure 2. The depth camera (RealSense L515, Intel Corporation, Santa Clara, CA, USA) was used to acquire NIR and depth images. The TIR camera (IRay P2, IRay Technology Co., Ltd., Yantai, China) was used to acquire TIR images. The two cameras were connected to an industrial computer (ARK-3531, Advantech Co., Ltd., Kunshan, China) via a USB interface, and the image acquisition and storage were controlled using the Python programming language.

The cameras were fixed on a self-developed mobile chassis via a ball head, a square hollow, a clamp, and a steel pipe. The height of the clamp was adjustable to capture images of chickens on different tiers, and the length of the square hollow and the angle of the ball head were adjustable to change the linear field of view of the cameras. The self-developed mobile chassis was controlled based on the robot operating system (ROS) with a combination of Python and C++ programming languages. The chassis was guided by magnetic tape to stop at fixed locations, and then the industrial computer sent a command to the cameras to collect images. Due to the small number of dead chickens, the dead chicken images were collected manually using the cameras and a tripod after locating the dead chickens by manual inspection.

#### 2.1.3. Image Data

On 10 and 11 November 2022, a total of 2052 sets of live chicken images (excluding dead chickens) were collected from the upper, middle, and lower tiers. From 11 November to 4 December 2022, 81 dead chickens were manually inspected, and a total of 1937 sets of dead chicken images (including dead chickens) were collected from different angles and distances. Each set of images included one TIR image, one NIR image, and one depth image, with resolutions of 256 × 192, 640 × 480, and 640 × 480, respectively, as shown in Figure 3.

### 2.2. Image Registration Method

Due to the differences in perspective, field of view, and resolution between TIR and depth cameras, image registration must be performed to align multiple images to ensure that temperature, texture, and depth information are correctly matched. This paper proposed a pixel-level registration method for TIR, NIR, and depth images based on coordinate transformation using depth information obtained from the depth camera.

#### 2.2.1. Image Registration Principle

Because the TIR camera had a smaller field than the depth camera, in order to retain more information after registration, the NIR image and depth image coordinates should be projected onto the NIR image coordinates. Since the relative positions of the TIR camera and the depth camera were fixed, the intrinsic and extrinsic parameters of the two cameras could be calculated under the same world coordinate system. After incorporating the depth information, the NIR and depth image coordinates were projected onto the TIR image coordinates. The registration process is illustrated in Figure 4, using the example of projecting the NIR image coordinates onto the TIR image coordinates.

(1)Transform the NIR image coordinates to the NIR camera coordinate system.

Based on the pinhole imaging model and the principle of similar triangles, the four parameters of the camera model were obtained, as shown in Equation (1).
(1)$${\left[\begin{array}{c} U\\ v\\ 1 \end{array}\right]}_{NIR}={\left[\begin{array}{ccc} k_{x} & 0 & u_{0}\\ 0 & k_{y} & v_{0}\\ 0 & 0 & 1 \end{array}\right]}_{NIR}{\left[\begin{array}{c} X/Z\\ Y/Z\\ 1 \end{array}\right]}_{NIR}$$
where ${\left[u,v\right]}_{NIR}$ is the NIR image coordinates, ${\left[k_{x},k_{y},u_{0},v_{0}\right]}_{NIR}$ is the intrinsic parameters of the NIR camera, and ${\left[X,Y,Z\right]}_{NIR}$ is the coordinate in the NIR camera coordinate system.

Equations (2) and (3) were derived from Equation (1), which expressed the coordinates in the NIR camera coordinate system.
(2)$$X_{NIR}=\frac{(u_{NIR}-u_{0NIR})Z_{NIR}}{k_{x}}$$
(3)$$Y_{NIR}=\frac{(v_{NIR}-v_{0NIR})Z_{NIR}}{k_{y}}$$
where $Z_{NIR}$ is depth information corresponding to the depth image.

(2)Transform the NIR camera coordinate system coordinates to the TIR camera coordinate system.

The transformation between the NIR camera coordinate system and the world coordinate system is shown in Equation (4). The transformation between the TIR camera coordinate system and the world coordinate system is shown in Equation (5).
(4)$${\left[\begin{array}{c} X\\ Y\\ Z \end{array}\right]}_{NIR}=R_{NIR}{\left[\begin{array}{c} X\\ Y\\ Z \end{array}\right]}_{W}+p_{NIR}$$
(5)$${\left[\begin{array}{c} X\\ Y\\ Z \end{array}\right]}_{TIR}=R_{TIR}{\left[\begin{array}{c} X\\ Y\\ Z \end{array}\right]}_{W}+p_{TIR}$$
where ${\left[X,Y,Z\right]}_{NIR}$ is the coordinate in the NIR camera coordinate system, ${\left[X,Y,Z\right]}_{W}$ is the coordinate in the world coordinate system, ${\left[X,Y,Z\right]}_{TIR}$ is the coordinates in the TIR camera coordinate system, $R_{NIR}$ is the rotation matrix of the NIR camera, $R_{TIR}$ is the rotation matrix of the TIR camera, $p_{NIR}$ is the translation vector of the NIR camera, and $p_{TIR}$ is the translation vector of the TIR camera. In the same world coordinate system, where ${\left[X,Y,Z\right]}_{W}$ is the same.

Equation (6) was derived from Equations (4) and (5). The coordinates in the NIR camera coordinate system were transformed into the coordinates in the TIR camera coordinate system using Equation (6).
(6)$${\left[\begin{array}{c} X\\ Y\\ Z \end{array}\right]}_{TIR}=R_{TIR}\left(R_{NIR}^{-1}\left({\left[\begin{array}{c} X\\ Y\\ Z \end{array}\right]}_{NIR}-p_{NIR}\right)\right)+p_{TIR}$$

(3)Transform the TIR camera coordinate system coordinates to the TIR image coordinates.

Similar to (1), coordinates in the TIR camera coordinate system were transformed into the TIR image coordinates using Equation (7).
(7)$${\left[\begin{array}{c} U\\ v\\ 1 \end{array}\right]}_{TIR}={\left[\begin{array}{ccc} k_{x} & 0 & u_{0}\\ 0 & k_{y} & v_{0}\\ 0 & 0 & 1 \end{array}\right]}_{TIR}{\left[\begin{array}{c} X/Z\\ Y/Z\\ 1 \end{array}\right]}_{TIR}$$
where ${\left[u,v\right]}_{TIR}$ is the TIR image coordinates, ${\left[k_{x},k_{y},u_{0},v_{0}\right]}_{TIR}$ is the intrinsic parameters of the TIR camera, and ${\left[X,Y,Z\right]}_{TIR}$ is the coordinates in the TIR camera coordinate system.

The transformation from the NIR image coordinates to the TIR image coordinates was achieved. The same steps could be used to transform the depth image coordinates to the TIR image coordinates. The registration process described above was implemented using Python and the OpenCV-Python library.

#### 2.2.2. Camera Intrinsic and Extrinsic Parameters Calibration

To obtain the intrinsic and extrinsic parameters of both cameras in the same world coordinate system, we made a calibration board that worked for both the TIR and depth cameras, as shown in Figure 5a. The calibration board was made of a 1 mm thick 7075 aluminum plate with six 5 cm square grids, then painted white. A heating plate covered with black tape was placed underneath the calibration board to build a checkerboard. Twenty-seven sets of calibration board images were captured from different angles and distances using both cameras, as shown in Figure 5b,c. The Zhang method [25] was used for calibration with MATLAB 2016a (MathWorks, Natick, MA, USA) and the calibration toolbox developed by Bouguet, J.-Y. [26].

### 2.3. Dataset

The registered TIR, NIR, and depth images were stacked in the TND image according to the RGB color space, where the R channel was the TIR image, the G channel was the NIR image, and the B channel was the depth image. Single-source and dual-source images were combinations derived from different channels of TND images. A sample set of 1250 TND images were manually selected, including 950 dead chicken images and 300 live chicken images. An image annotation tool named Labelimg was used to select the chickens’ contours and label them as dead or live. We randomly selected 60% of the TND images for model training, 20% for validating the model’s performance and adjusting the model training parameters, and 20% for evaluating the model’s generalization ability.

### 2.4. Dead Chicken Detection Network

YOLOv8n, Deformable DERT, Cascade R-CNN, and TOOD were selected as the dead chicken detection networks. The performance of dead chicken detection using single-source, dual-source, and multi-source images was explored by changing the channels of the TND image.

#### 2.4.1. YOLOv8n

The YOLO series is a typical one-stage object detection algorithm. YOLOv8 is the latest YOLO series algorithm developed by the YOLOv5 team. The algorithm replaces the C3 module in the YOLOv5 backbone network with C2f, introduces a new Anchor-Free detection head, and a new loss function. According to the official test results on the COCO Val 2017 dataset, although YOLOv8 has a corresponding increase in model parameter size and FLOPs compared to YOLOv5, it has significantly improved accuracy and is a state-of-the-art model. YOLOv8n is a network with the fewest network layers in YOLOv8.

#### 2.4.2. Deformable DETR

Deformable DETR [27] is a representative algorithm that applies Transformer neural network architecture to object detection. In response to issues such as slow convergence speed and limited spatial resolution of DETR [28], Deformable DETR introduces a deformable attention mechanism to DETR, accelerating its convergence speed and improving its detection performance for small objects.

#### 2.4.3. Cascade R-CNN

Cascade R-CNN (Cai et al., 2017) [29] is one of the representatives of two-stage object detection algorithms. To address the issues that most of the candidate boxes selected by the region proposal network have low quality at low IoU and simply increasing the IoU threshold may lead to overfitting and mismatching problems, Cascade R-CNN proposes a multi-detector structure. It uses the output of the previous stage detector to train the next stage detector and adopts higher IoU thresholds for each subsequent stage to generate higher-quality rectangular boxes.

#### 2.4.4. TOOD

TOOD (Feng et al., 2021) [30] is a task-aligned one-stage object detection algorithm. The traditional one-stage object detection algorithms use two parallel branches for classification and localization tasks. This might lead to a certain level of spatial misalignment in predictions between the two tasks. TOOD designed a task-aligned head (T-Head) to increase the interaction between the two tasks and proposed task alignment learning (TAL) to explicitly pull (or even unify) the optimal anchors closer for the two tasks.

We implemented YOLOv8n based on the code provided by the authors on GitHub [https://github.com/ultralytics/ultralytics, accessed on 1 March 2023]. We implemented Deformable DETR, Cascade R-CNN, and TOOD based on the MMDetection deep learning object detection toolkit, with ResNet50 as the backbone network. All other parameters were set to default except for changing the training epochs for Deformable DETR, Cascade R-CNN, and TOOD to 50, 36, and 72.

### 2.5. Computing Resource

The dead chicken detection networks’ training, validation, and testing were based on the PyTorch deep learning framework, using the Python programming language and GPU acceleration for computing. The experiments were conducted on a server running Ubuntu 18.04 with an Intel(R) Xeon(R) Gold 6133 CPU @ 2.5 GHz, 40 GB of RAM, and a Tesla V100 SXM2 32 GB GPU. (To facilitate the reproducibility of experiments, it is recommended to use a graphics processing unit (GPU) with a memory size of 12 GB or higher).

### 2.6. Evaluation Metrics of Object Detection Networks

To evaluate the dead chicken detection performance of different types of images, AP50, R, AP75, and AP@50:5:95 were used as the evaluation metrics. The R and AP formulas are shown in Equations (8)–(10).
(8)$$P=\frac{TP}{TP+FP}$$
(9)$$R=\frac{TP}{TP+FN}$$
(10)$$AP=\int_{0}^{1}P(R)dR$$
where TP is the number of positive samples that are correctly predicted, FP is the number of positive samples that are incorrectly predicted, and FN is the number of negative samples that are incorrectly predicted. AP is the detection accuracy of a single category. AP50 and AP75 are the detection accuracy when IoU = 0.5 and IoU = 0.75, respectively. AP@50:5:95 is the mean detection accuracy when IoU is set to 0.5, 0.55, 0.6, 0.65, 0.7, 0.75, 0.8, 0.85, 0.9, and 0.95 (IoU = 0.5:0.95).

## 3. Results and Discussion

### 3.1. Registration Result and Discussion

The registered TIR, NIR, and depth images are shown in Figure 6a–c. After registration, many black noise points were observed in the NIR and depth image. Two factors caused these noises. First, the raw depth image contained black noise points, resulting in the loss of some image information when depth information was used for registration. This factor was the reason for generating streak noise near the chicken cage. Second, the TIR and depth cameras had different fields of view and were in different positions, so the directions of image acquisition were biased, resulting in some information loss when the NIR and depth images were projected onto the TIR image. This phenomenon was similar to the human eyes acquiring different directional information about the same object. This factor caused black noise near the chicken contour and was not avoidable. However, the number of black noise points near the chicken contour could be reduced by decreasing the distance between the depth and TIR cameras. Since the NIR and depth images were projected onto the TIR image, and the TIR image did not undergo a coordinate transformation, there was no black noise in the TIR image.

The TND image stacked by the registered TIR, NIR, and depth images was shown in RGB color space as in Figure 6d. Observing the chicken cage below the water pipe in the TND image, we could find an apparent misalignment caused by the lack of depth information and the different camera angles mentioned above. Observing the chicken’s head, body contour, feet, and the water pipe in the TND image, we could conclude that the three channels overlapped well, indicating that the proposed registration method using depth information was practical and high-quality.

### 3.2. Detection Results

The dead chicken detection results of the single-source, dual-source, and multi-source images in YOLOv8, Deformable DERT, Cascade R-CNN, and TOOD are shown in Figure 7.

#### 3.2.1. The Detection Results of Single-Source Image

Figure 7 shows the metrics of AP50, R, AP75, and AP@50:5:95 with the TIR, NIR, and Depth images. According to Figure 7a,b, when the IoU threshold was set to 0.5, the detection performance of models with single-source images in different networks was not entirely the same. The model with the NIR image achieved the best performance in YOLOv8n, Deformable DETR, and Cascade R-CNN. The model with the depth image had the best detection performance in TOOD. The model with the TIR image had the worst detection performance. Among all the detection results with single-source images, the model with the NIR image had the best dead chicken detection performance in Deformable DETR, with an average precision (AP) of 98.9% (IoU = 0.5) and a recall of 98.5% (IoU = 0.5). From Figure 7c,d, with the increase of IoU, the detection performance of models with single-source images tended to be consistent. The model with the NIR image had the best detection performance of dead chickens, followed by the depth and TIR image. The AP of the model with the NIR image was 89.5% (IoU = 0.75) and 74.4% (IoU = 0.5:0.95) in Deformable DETR.

Some detection results with single-source images are shown in Figure 8. Many problems existed in the results with the TIR image, such as missing detection, false detection, repeated detection, and inaccurate detection box. The missing detection was shown in Figure 8c, possibly because the chickens were not dead for very long and had similar temperature characteristics as the live chickens. The false detection is shown in Figure 8d,e. The tails of live chickens and troughs in the TIR image could easily be misconstrued as belonging to dead chickens, which might be because the temperature of the tails and troughs was low, similar to those of dead chickens. The repeated detection was shown in Figure 8b, possibly because the dead chicken and live chicken outlines were connected, and the temperature characteristics were similar. The inaccurate prediction box is shown in Figure 8a,e. The dead chicken prediction boxes were included in the annotation boxes, which might be because the chicken’s body temperature gradually decreases after death, and the outline of the dead chicken was not evident in the TIR image.

By analyzing the detection results with the NIR images, it was found that almost all models with the NIR image accurately located the position of dead chickens, and there was no missing detection. However, these models still had repeated and false detection problems. The repeated detection was shown in Figure 8d–f, which might be because live chickens or chicken cages separated the outline of dead chickens, similar to the presence of multiple dead chickens. The false detection is shown in Figure 8d. The live chicken’s tail was misidentified as being a part of the dead chicken, which might be because the chickens were seriously overlapped in the image, and the tail of a live chicken was easily misconstrued as the tail of a nearby dead chicken. 

By analyzing the detection results with the depth image, it was found that the problems of inaccurate detection box, missing detection, and repeated detection. The inaccuracy and missing detection boxes were shown in Figure 8a,f, possibly because the depth image only contained distance information. When the distance features of the dead chicken were similar to those of the live chicken or the dead chicken was connected to the live chicken, the object detection algorithm had difficulty detecting the chicken accurately. The repeated detection was shown in Figure 8d, similar to that in the NIR image. 

From Figure 8a–c, it was found that the detection performance with the NIR image was better than that with the TIR image and depth image. This finding was consistent with the detection results of Figure 7, which might be because the NIR image contained complete outline information of the dead chicken. Consequently, when using a single-source image for dead chicken detection, the NIR image could help achieve higher detection accuracy than the TIR and depth images.

#### 3.2.2. The Detection Results of Dual-Source Image

Figure 7 shows the metrics of AP50, R, AP75, and AP@50:5:95 of the TIR-NIR, TIR-depth, and NIR-depth images. According to Figure 7a,b, when the IoU threshold was set to 0.5, the detection performance of the model with the NIR-depth image was the best, with an AP of 99.7% (IoU = 0.5), and a recall of 99.0% (IoU = 0.5) in Deformable DETR. In Cascade R-CNN and TOOD, the detection performance of the model with the TIR-NIR image or the NIR-depth image was significantly better than that with the TIR-depth image. Compared with the models with single-source images, the performance of partial models with dual-source images was improved. For example, the model with the NIR-depth image in Deformable DETR and Cascade R-CNN performed better than that with single-source images. However, the performance of some models with dual-source images was not significantly improved or even decreased compared to those with single-source images. For example, the detection performance of the model with the TIR-depth image in Cascade R-CNN and TOOD decreased compared with the model with single-source images. From Figure 7c,d, with the increase of IoU threshold, the detection performance of the models with dual-source images was similar, and the dead chicken detection performance of the models with the TIR-NIR or NIR-depth image was better than that with the TIR-depth image. Compared to the models with single-source images, the AP of the models with dual-source images showed an improvement. In all detection results with dual-source images, the AP of the model with the NIR-depth image was 91.1% (IoU = 0.75), and that with the TIR-NIR image was 76.3% (IoU = 0.5:0.95) in Deformable DETR.

Partial detection results with dual-source images are shown in Figure 9. The detection performance of the models with the TIR-NIR or the NIR-depth image was better than that with the TIR-depth image in most models, as shown in Figure 9a,e. This finding was consistent with the results from Figure 7, which might be because the NIR image contained clear outlines of the dead chicken. In contrast, the TIR and depth images lacked apparent outline features of the dead chickens. 

Compared with the results with single-source images, the number of false and repeated detection boxes in dual-source images was significantly reduced. For example, the false detection boxes in Figure 8d,e disappeared compared with the detection results in Figure 9d,e. The repeated detection boxes of the TIR image in Figure 8b disappeared compared with the detection results of the TIR-depth image in Figure 9b. This result might be because one channel in the dual-source images could accurately identify dead and live chickens, making up for the deficiency of the other channel in detecting chickens. In addition, it should be noted that not all models showed improved detection performance compared to models with single-source images. The detection performance of the model with the NIR-depth image in Figure 9a was lower than that with the NIR image in Figure 8a, and the model with the TIR-depth image in Figure 9a failed to even detect dead chickens. This result might be caused by the fact that the dead chickens shared similar features with live chickens in one channel and decreased the overall detection performance. It indicated that an increase in the number of source images did not necessarily help improve the dead chicken detection performance, which aligned with the conclusion drawn from the analysis of Figure 7. 

In conclusion, the models with dual-source images could reduce the number of false and repeated detection boxes compared to those with single-source images. When dual-source images were used for dead chicken detection, the TIR-NIR and NIR-depth image could perform better than the TIR-depth image.

#### 3.2.3. The Detection Results of Multi-Source Image

Figure 7 shows the metrics of AP50, R, AP75, and AP@50:5:95 with the TIR-NIR-depth image. According to Figure 7a,b, when the IoU threshold was set to 0.5, the model’s performance with the multi-source image was not significantly improved compared with the models with single-source or dual-source images. In fact, in some networks, the detection performance with the multi-source image was even decreased. For example, in Deformable DETR and Cascade R-CNN, the detection performance of the model with the multi-source image was lower than that with the NIR-depth image. In TOOD, the detection performance of the model with the multi-source image was lower than that with the NIR and the depth image. Among all detection results where the IoU threshold was set to 0.5, the model with the multi-source image in YOLOv8n had the best detection performance, with an AP of 98.6% (IoU = 0.5) and a recall of 97.3% (IoU = 0.5). From Figure 7c,d, with the increase of IoU, the detection performance of the model with the multi-source image was significantly improved compared with the model with single-source images and the TIR-depth image. However, it remained unchanged or slightly improved compared with the model with the TIR-NIR or NIR-depth images. Among all the detection results where the IoU threshold was set to higher than 0.5, the model with the multi-source image in Deformable DETR had the best detection performance, with an AP of 91.0% (IoU = 0.75) and 76.7% (IoU = 0.5:0.95).

Partial detection results with the multi-source image are shown in Figure 10. Most prediction boxes in the multi-source image perfectly overlapped with the annotation boxes, as shown in Figure 10b,c,e. This indicated that the detection performance of the model with the multi-source image was satisfactory. Compared with the detection results of the models with single-source or dual-source images in Figure 8 and Figure 9, the number of missing, false, and repeated detection boxes in the multi-source image was significantly reduced. This indicated that the detection performance of the model with the multi-source image was better than that with single-source and dual-source images. Meanwhile, the detection boxes’ accuracy of the model with the multi-source image was slightly improved compared with that with dual-source images. For example, the accuracy of the prediction box in Figure 10d was better than that of the TIR-NIR and TIR-depth images in Figure 9d. Nevertheless, the detection performance of a few models with multi-source images was not better or worse than that with dual-source and single-source images. As shown in Figure 10f, the prediction box was almost identical to the TIR-depth and the NIR-depth image in Figure 9f. As shown in Figure 10a, the prediction box only contained a part of the dead chicken, while the NIR image in Figure 8a could locate the dead chicken more accurately. This indicated that the detection performance of the model with the multi-source image was easily affected by a single channel that made up the multi-source image, resulting in inconspicuous improvement in detection performance. 

In summary, the model with the multi-image could achieve high detection performance. Compared with single-source or dual-source images, the model with the multi-source image could reduce the number of false detection, missed detection, and repeated detection boxes. The detection performance of the model with the multi-source image was significantly improved compared with the model with single-source images and the TIR-depth image and slightly improved compared with the model with the TIR-NIR and NIR-depth images.

### 3.3. Limitations and Future Works

As mentioned in Section 3.1, the image registration method generated black noise, which could potentially diminish the accuracy of the dead chicken detection networks. Moreover, this registration method was time-consuming and not suitable for real-time detection by inspection robots. Furthermore, the requirement of a depth camera for this image registration method added to the expenses and posed challenges to its widespread adoption. Finally, high-specification graphics cards were not suitable for inspection robots due to their high power consumption.

To expedite image registration, our next step is to further simplify the process using the C++ programming language. Additionally, we are considering employing deep learning to achieve image registration and fusion in the future. In order to achieve real-time dead chicken detection with inspection robots, our next objective is to enhance the YOLOv8 network. This enhancement will prioritize optimizing detection speed while maintaining accuracy. Subsequently, we will deploy the network to embedded systems such as NVIDIA Jetson to reduce power consumption. Ultimately, we aim to integrate these embedded systems with our self-developed mobile chassis for field testing on commercial farms.

## 4. Conclusions

In this study, we first proposed a pixel-level registration method using depth information to align TIR, NIR, and depth images and projected the NIR and depth images into the TIR image coordinate system to achieve registration. To evaluate the performance with different source images, we trained dead chicken detection models with the registered single-source, dual-source, and multi-source images using representative object detection networks, including YOLOv8, Cascade R-CNN, TOOD, and Deformable DERT. At an IoU threshold of 0.5, the detection performance with each image was not identical. As the IoU threshold increases, the detection results show a similar trend: the model with the NIR image performed best among models with single-source images, and the models with dual-source images performed better than that with single-source images. The model with the TIR-NIR image or the NIR-depth image performed better than the model with the TIR-depth image. The detection performance with the multi-source images was better than that with single-source images but was not significantly different from that with the TIR-NIR and NIR-depth images. Therefore, the TIR-NIR image could be used for dead chicken detection to achieve high accuracy and reduce the cost. 

Overall, we proposed a high-accuracy method for detecting dead chickens on commercial farms that was robust to ambient light. We also evaluated the performance of various dead chicken detection models using different source images. Our findings may prove to be useful for future research on poultry health monitoring using near-infrared, thermal infrared, or depth cameras.

## Figures and Tables

**Figure 1 animals-13-01861-f001:**
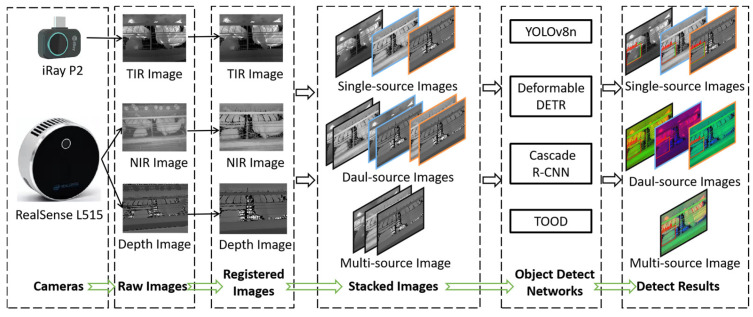
Flow diagram.

**Figure 2 animals-13-01861-f002:**
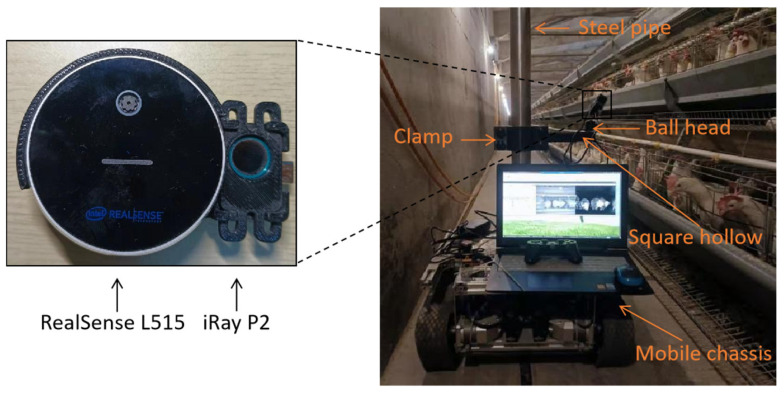
Image acquisition device.

**Figure 3 animals-13-01861-f003:**
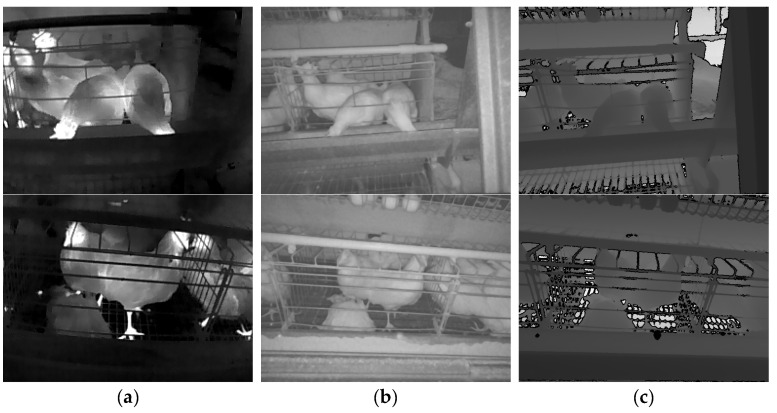
Raw images. (**a**) TIR image. (**b**) NIR image. (**c**) Depth image. Note: The images in the first row do not contain any dead chickens, while the images in the second row contain a dead chicken.

**Figure 4 animals-13-01861-f004:**
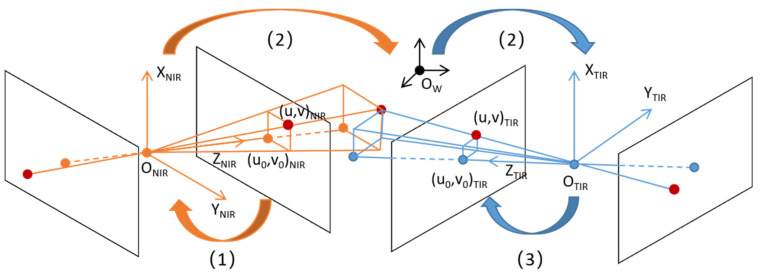
Flow of coordinate transform. Note: The meaning of variables and coordinate systems in Figure 4 was described in the following transformation steps.

**Figure 5 animals-13-01861-f005:**
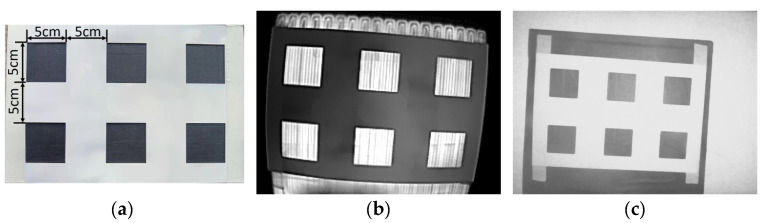
Calibration board. (**a**) Photograph. (**b**) TIR image. (**c**) NIR image.

**Figure 6 animals-13-01861-f006:**
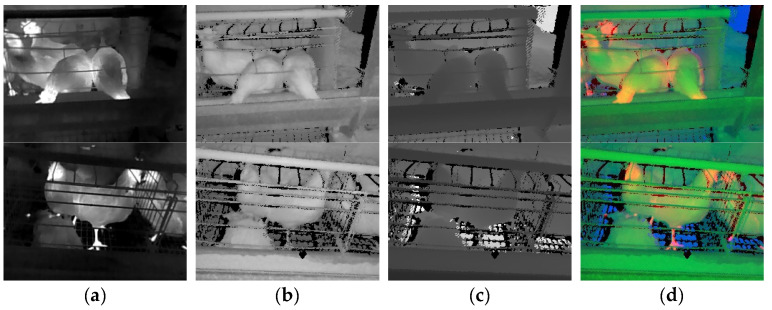
Registered images. (**a**) TIR image. (**b**) NIR image. (**c**) Depth image. (**d**) TND image. Note: The first row is live chicken images, and the second is dead chicken images.

**Figure 7 animals-13-01861-f007:**
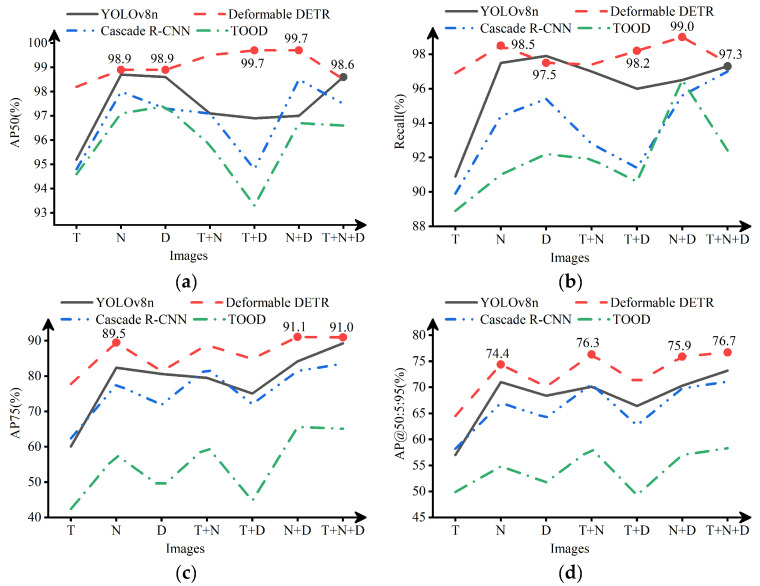
Results of dead chicken detection models. (**a**) AP50. (**b**) Recall. (**c**) AP75. (**d**) AP@50:5:95. Note: T represents the TIR image, N represents the NIR image, D represents the depth image, T + N represents the TIR-NIR image, T + D represents the TIR-depth image, N + D represents the NIR-depth image, and T + N + D represents the TIR-NIR-depth image.

**Figure 8 animals-13-01861-f008:**
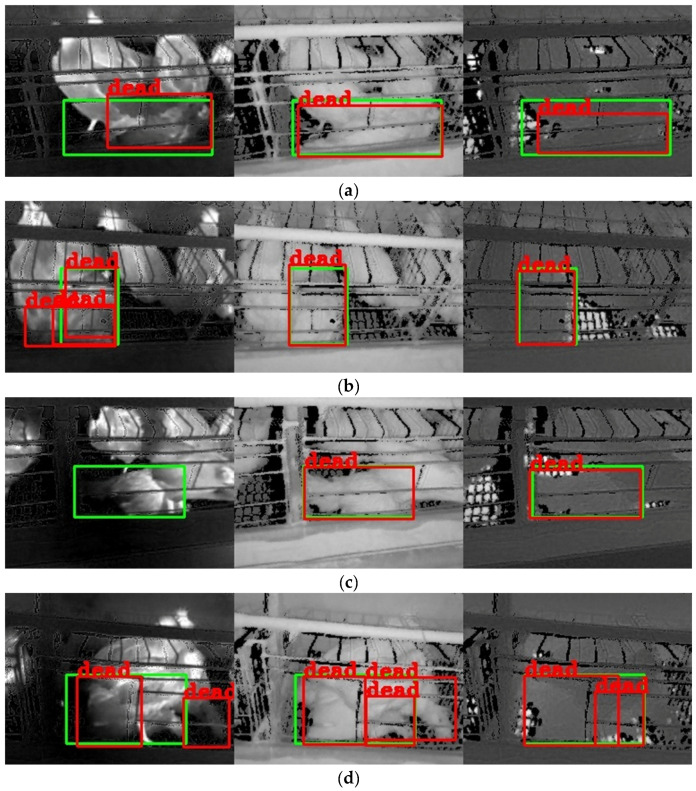
Detection results with single-source images. Note: (**a**–**f**) are mosaic images. From left to right, the images are TIR, NIR, and depth images in a mosaic image. The green boxes in the image are the annotation boxes, and the red boxes are the prediction boxes of the YOLOv8n object detection algorithm. All bounding boxes for live chickens were removed to enable clear observation of the dead chicken detection.

**Figure 9 animals-13-01861-f009:**
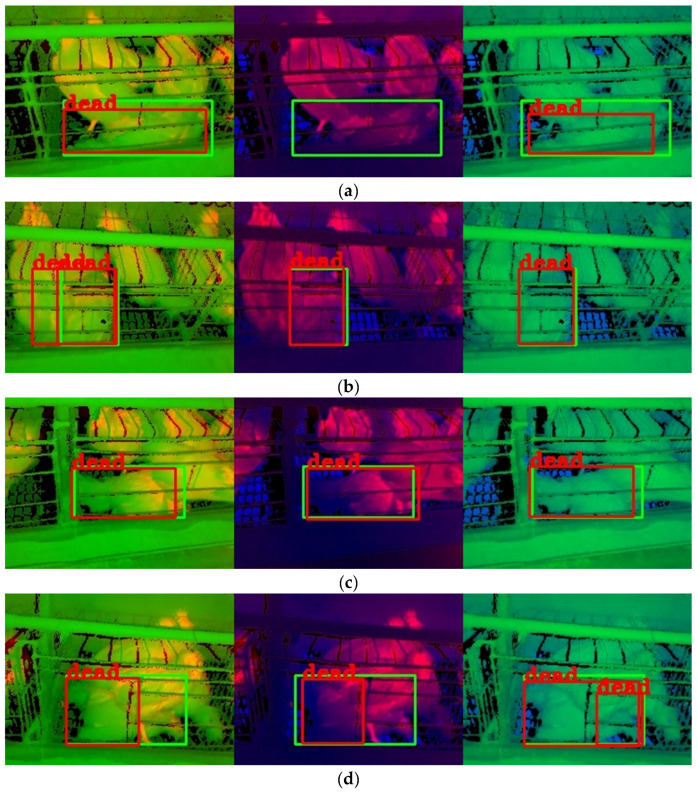
Detection results with dual-source images. Note: (**a**–**f**) are mosaic images. From left to right, the images are TIR-NIR, TIR-depth, and NIR-depth images in a mosaic image. TIR, NIR, and depth images correspond to the R, G, and B channels of RGB color space, respectively. The idle channel in the dual-source image is set to zero. The green boxes in the image are the annotation boxes, and the red boxes are the prediction boxes of the YOLOv8n object detection algorithm. All bounding boxes for live chickens were removed to enable clear observation of the dead chicken detection.

**Figure 10 animals-13-01861-f010:**
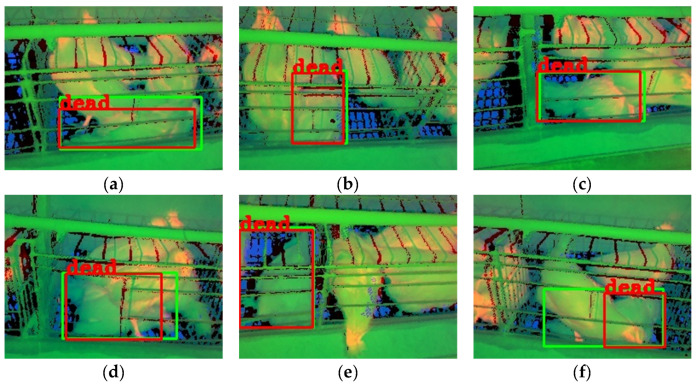
Detection results with the multi-source image. Note: (**a**–**f**) are TND images. TIR, NIR, and depth images correspond to the R, G, and B channels of RGB color space, respectively. The green boxes in the image are the annotation boxes, and the red boxes are the prediction boxes of the YOLOv8n object detection algorithm. All bounding boxes for live chickens were removed to enable clear observation of the dead chicken detection.

## Data Availability

The data are not publicly available due to privacy reasons.

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
