# Peer review of "Dead Laying Hens Detection Using TIR-NIR-Depth Images and Deep Learning on a Commercial Farm"

_animals, 2023, doi:10.3390/ani13111861_

Round 1
Reviewer 1 Report
The scientific term "IoU" was frequently shown in the manuscript. But authors did not provide the definition of this term. The IoU term would be found in different way (such as IOU) to be presented in the manuscript. Authors should keep them consistently, and make the definition of it much clearer.
The methods to apply the IoU(0.5) and IoU(0.5:0.95) to the analysis could not be recognized in the manuscript. Readers could not easily figure out how to define the parameters in the IoU term.
The TND image could not provide the definition to identify the image. Authors should provide the whole scientific term to present the abbreviation TND.
There are so many parameters, abbreviations, and scientific terms in the manuscript. Authors should provide a paragraph to introduce the terms that would be found in the article.
Actually, the research was fine and useful to the poultry industry. Hope authors could provide the information well to publish the article.
Reviewer 2 Report
The article sent for evaluation brings an important contribution and potential application in breeding environments.
I believe that the structure of the document was very well organized and allowed a great understanding of the contents addressed.
The digital image processing techniques applied jointly allow for a glimpse of the potential for improving and refining the responses, with a high percentage of correct responses being achieved in the detection of dead birds.
I missed the speed indications in the article for processing in each of the modalities tested and compared. The same can be said regarding the computational cost of each of these experimented algorithms.
Minimum hardware considerations for experiment replication would also be most welcome.
Furthermore, the text is very well written and the title represents the research carried out.
The introduction and justification make it clear to readers the importance of the topic and provide a good foundation for previous works already carried out.
The way chosen for presenting the materials and methods was, in my view, very interesting, segmenting the data collection with the image acquisition devices, presenting below the data obtained from the images in TIR, NIR, and Depth mode.
Afterward, he addressed the way of recording the images, where the equations for modeling seem correct to me, and the procedures for calibrating the images.
Next, he presented the Dataset and characteristics of the detection network using YOLOv8n, Deformable DETR, Cascade R-CNN, and TOOD.
In the sequence, the topic "Computing Resource" is presented, which I believe should be further explored with more information to facilitate the reproduction of the experiment, considering the minimum configurations to be used to meet and achieve results similar to those obtained in this experiment.
A strong point of the work was the Evaluation metrics of object detection networks, which, in my opinion, contributed a lot to the quality of the manuscript.
The results and discussions section is also very complete and very clearly brought a great discussion.
I feel that the possibility of a greater comparison of this advance obtained in this experiment with those obtained by the researchers mentioned in the introduction could have been further explored. I believe that this inclusion could add more value to the work.
The graphics and images are of excellent quality and allow a good understanding of the steps explored and procedures covered.
I consider the approach excellent with consideration of the errors obtained during experimentation since the detection of dead birds in this environment of extensive production in cages is very complex.
It was also very convenient to indicate proposals for future work, where the authors mention seeking to improve accuracy and speed (which I observe should be better addressed in the manuscript for improvement and understanding of the hardware requirements for reproducing the experiment by readers interested in the proposal methodological).
The conclusions are very consistent and reinforce that the proposed methodology led to the fulfillment of the proposed objectives and allowed for achieving good accuracy.
I congratulate the authors and appreciate the opportunity to review the manuscript.
Reviewer 3 Report
In this paper, the authors present a novel method for dead chicken detection based on the use of multi-source images and deep learning.
The document is mostly well written, without relevant English language issues. It is organized as follows: Simple Summary, Abstract, a first Section containing an Introduction, a second Section on Materials and Methods, a third Section on Results and discussion, a fourth Section with the Conclusion of the work, and finally the References used.
After a thorough review, I believe that the paper could be of some interest to the readers of the journal Animals. However, I have some questions and suggestions for the Authors. Please, see my comments below.
1. Has anybody previously used multi-source images for chicken dead detection? Please, expand also information given in Lines 109-114, related to the use of multi-source images in the livestock industry.
2. In relation to multi-source images, I consider that the Authors should clarify how is it possible to deal with them in the Introduction. Are they processed one at a time, or is there some kind of information fusion process? I recommend to explore and detail these aspects.
3. The information provided about the image acquisition rig in Figure 1 should be extended to for a better understanding of the device, its composition and expected features.
4. In relation with the information mentioned in Lines 154-156, would it be possible to obtain the same information with the robot? Additionally, are the characteristics of those images obtained with a tripod similar to those obtained with the robot?
5. In section 2.2.1., the Authors should explain the reason why it is necessary the transformation from the NIR image coordinates to the TIR image coordinates.
6. The dataset used seems to be imbalanced, how can this affect to the performance of the model?
7. What were the criteria for the selection of the neural networks used? Why these architectures and no others?
8. What is the core theoretical contribution of the paper?
9. I recommend splitting Results and Discussion into two different sections.
10. It could be useful to add a flow diagram explaining how the information is transformed from the moment it is obtained with the cameras, until a prediction is given. I consider that this could be very useful to improve the paper and help the readers to understand in a visual way the proposal made.
11. In comparison with other proposals of the state of the art, why is this better than other options? What are its benefits and weaknesses? What is the relevance of the proposal in the field of study?
12. What were the limitations of this work?
13. What could be the costs of implementing a system like this is into a real farming industrial setting? Would it be possible to use the system in any farm before a prior training process? How could this be done?
14. The paper needs to extend itself more about a vision of future related works.
15. Other comments
• I consider that it could be useful to add a paragraph at the end of the Introduction section explaining how the manuscript is organized.
• Revise and remove Lines 296–298,as they might seem to belong to from the template.
• It would be very useful to add a list of abbreviations at the end of the paper.
I recommend revising the figure titles regarding their grammar composition. In a few cases, starting with 'The' would not be the better option for a label, for example in Figures 3 and 6, among others.
